# Estimated Glomerular Filtration Rate and Hearing Impairment in Japan: A Longitudinal Analysis Using Large-Scale Occupational Health Check-Up Data

**DOI:** 10.3390/ijerph191912368

**Published:** 2022-09-28

**Authors:** Hiroshi Miyake, Takehiro Michikawa, Satsue Nagahama, Keiko Asakura, Yuji Nishiwaki

**Affiliations:** 1Department of Environmental and Occupational Health, Toho University Graduate School of Medicine, 5-21-16, Omorinishi, Otaku, Tokyo 143-8540, Japan; 2Department of Environmental and Occupational Health, School of Medicine, Toho University, 5-21-16, Omorinishi, Otaku, Tokyo 143-8540, Japan; 3Division of Occupational Health and Promotion, All Japan Labor Welfare Foundation, 6-16-11, Hatanodai, Shinagawa, Tokyo 142-0064, Japan

**Keywords:** estimated glomerular filtration rate, follow-up, hearing impairment, occupational health, serum creatinine

## Abstract

Several longitudinal studies have examined associations between renal dysfunction and hearing impairment. Here, we explored the longitudinal association between estimated glomerular filtration rate (eGFR) and hearing impairment among the working-age population in Japan. Participants were 88,425 males and 38,722 females aged 20–59 years, without hearing impairment at baseline (2013), who attended Japanese occupational annual health check-ups from 2013 to 2020 fiscal year. eGFR was categorized into four groups (eGFR upper half of ≥90, lower half of ≥90 (reference), 60–89, and <60 mL/min/1.73 m^2^). Low- and high-frequency hearing impairment were assessed using data from pure-tone audiometric testing. A Cox proportional hazards model was applied to estimate hazard ratio (HR) values for hearing impairment. Low eGFR did not increase the risk of low- or high-frequency hearing impairment. For males, multivariable-adjusted HR of high-frequency hearing impairment was 1.16 (95% confidence interval, 1.01–1.34) for the upper half of the ≥90 mL/min/1.73 m^2^; however, this positive association between high eGFR and high-frequency hearing impairment did not appear to be robust in a number of sensitivity analyses. We conclude that, among the Japanese working-age population, eGFR was not generally associated with hearing impairment in people of either sex.

## 1. Introduction

Hearing impairment is among the most important disabilities that hampers quality of life, by interfering with communication and conversation; decreasing cognitive function; and contributing to depression, loss of education, and employment opportunities, as well as social isolation, loneliness, and stigma [1,2,3]. As of 2019, approximately 1.57 billion people, equating to one in five of the global population, had hearing impairment [4]. Risk factors for hearing impairment fall into two major categories: congenital and acquired. Established acquired risk factors include several modifiable elements, such as loud noise, ototoxic medicines/chemicals, nutritional deficiencies, and viral infections. Additionally, there is evidence that lifestyle-related diseases, including cardiovascular disease, high blood pressure [5], obesity, atherosclerosis [6], and diabetes [7,8,9], are risk factors for hearing impairment. Chronic kidney disease (CKD), defined as kidney damage or glomerular filtration rate (GFR) < 60 mL/min/1.73 m^2^ for ≥3 months, regardless of cause, is a lifestyle-related disease and a risk factor for cardiovascular dysfunction [10]. Some cross-sectional studies have found a positive association between CKD and hearing impairment [11,12,13,14], and CKD may be among the risk factors for hearing impairment, in addition to cardiovascular disease.

Renal dysfunction, including end-stage renal disease, dialysis, or kidney transplantation, is associated with hearing impairment. As the stria vascularis of the renal glomerulus is physiologically and structurally similar to the cochlea, the disruption of ionic exchange characterizing renal dysfunction may also occur in the cochlea [15]. In the inner ear of uremic animals, reductions in Na^+^K^+^-activated ATPase activity, which is important for maintaining cation gradients in the cochlea, were observed [16]. In an experimental study using guinea pigs, artificial chronic renal failure due to nephrectomy led to changes in the threshold elevations of action potential and cochlear microphonics, which are indicators of cochlear dysfunction [17]. Thus, renal dysfunction is a potential cause of cochlear damage and may lead to a decline in hearing ability.

Here, we hypothesized that renal dysfunction could lead to hearing impairment. As few longitudinal studies have been conducted, there is insufficient evidence regarding whether renal dysfunction precedes hearing impairment. In Japan, a pure-tone audiometric test (30 dB at 1000 Hz and 40 dB at 4000 Hz) is a mandatory part of regular occupational health check-ups [18]. Therefore, we used data from occupational health check-ups to investigate the longitudinal association between estimated GFR (eGFR), as a marker of renal function, and hearing impairment among the working-age population in Japan.

## 2. Materials and Methods

### 2.1. Study Population

In Japan, according to the Industrial Safety and Health Act, employers are required to carry out occupational health check-up once a year for employees. Annual occupational health check-up data from Japanese workers were obtained from the All Japan Labor Welfare Foundation (Tokyo, Japan), which is a health service provider with several facilities in eastern Japan (Tokyo, Aomori, Nagano, Yamagata, Ibaraki, Gunma, and Nagoya prefectures). Data were anonymized and included information on all workers (20–59 years old) who had health check-ups in the 2013 fiscal year (April 2013 to March 2014) and agreed to the secondary use of their data. Measurement of serum creatinine to calculate eGFR is not mandatory for the health check-up but is conducted at the request of the employer or employee. Among 168,529 participants aged 20–59 years old who had serum creatinine data, those who did not have information on hearing test (*n* = 6858) or already had finding(s) on hearing test at baseline (*n* = 7046) were excluded. After a request for follow-up data from the All Japan Labor Welfare Foundation until March 2021, participants who visited only in 2013 (*n* = 27,478) were also excluded. Finally, a total of 127,147 participants (88,425 male and 38,722 female) were included in this study (Figure 1).

The study protocol was approved by the Ethics Committee of the Faculty of Medicine, Toho University (Approval No. A21087_A20011_A18095_A17107_A16130).

### 2.2. Evaluation of Hearing Impairment

Pure-tone air-conduction audiometry was conducted by trained staff using an audiometer (AA-57, RION, Tokyo, Japan) at the time of the annual health check-up. In accordance with the stipulations of Japan’s Industrial Safety and Health Act on health check-ups for workers, pure-tone hearing ability was tested with a signal intensity of 30 dB at a frequency of 1000 Hz, and with intensity of 40 dB at 4000 Hz. Low-frequency hearing impairment was defined as the inability to hear a pure-tone signal of 30 dB at 1000 Hz in the better ear, and high-frequency hearing impairment was defined as the inability to hear a pure-tone signal of 40 dB at 4000 Hz in the better ear [9]. The onset of hearing impairment was defined as the health check-up day on the first finding of hearing impairment and included cases where hearing impairment was apparent from conversation with the individual.

### 2.3. Evaluation of Renal Function by eGFR

Venous serum samples were collected and stored in a cooler at 4 °C for transportation to an external laboratory (SRL, Tokyo, Japan). Serum creatinine was analyzed using an enzymatic method. The reference values for serum creatinine used by the All Japan Labor Welfare Foundation are 0.61–1.04 mg/dL for males and 0.47–0.79 mg/dL for females. eGFR was calculated using revised equations for eGFR from serum creatinine in Japan [19], as follows:

eGFR (mL/min/1.73 m^2^) = 194 × serum creatinine (–1.094) × age (–0.287) [× 0.739 (if female)]

Participants were classified into six categories (eGFR ≥ 90, 60–89, 45–59, 30–44, 15–29, <15 mL/min/1.73 m^2^) using eGFR, according to the GFR classification of CKD severity by the Japanese Society of Nephrology [20]. As the study population was relatively young (mean age, 43.8 years), mean eGFR was high, and the eGFR ≥ 90 mL/min/1.73 m^2^ category included participants with a wide range of eGFR values; therefore, participants with eGFR ≥ 90 mL/min/1.73 m^2^ were dichotomized into two groups according to the sex-specific median values. Similarly, the four categories < 60 mL/min/1.73 m^2^ were summarized as one group because few participants had eGFR < 60 mL/min/1.73 m^2^. For the final analyses, participants were classified into four groups: eGFR upper half of ≥90, lower half of ≥90 (reference), 60–89, and <60 mL/min/1.73 m^2^.

### 2.4. Definition of Baseline Variables

Information on potential confounding factors that have been associated with hearing impairment [21,22,23,24,25,26] was collected, as follows: height, weight, blood pressure, triglyceride, high-density lipoprotein cholesterol, hemoglobin A1c (HbA1c) and hemoglobin levels in blood, and information on history of diabetes, medication, smoking status, alcohol consumption, exercise, and job type. Based on this information, we defined the following variables and used as the potential confounding factors. Body mass index (BMI) was calculated as weight in kilograms divided by height in meters squared and categorized into three groups: <18.5, 18.5–24.9, and ≥25 kg/m². Based on the definitions in NCEP-ATPⅢ (2005 revision) criteria for clinical diagnosis of metabolic syndrome [27] and Japanese criteria for metabolic syndrome [28], hypertension was defined as systolic blood pressure ≥ 130 mm Hg, diastolic blood pressure ≥ 85 mm Hg, or receiving medication, and dyslipidemia was defined as triglyceride level ≥ 150 mg/dL (1.7 mmol/L), high-density lipoprotein cholesterol level < 40 mg/dL (1.03 mmol/L) in males and <50 mg/dL (1.3 mmol/L) in females, or receiving medication. Diabetes was defined as self-reported diagnosis of diabetes, receiving medication, or HbA1c ≥ 6.5%. As HbA1c was obtained as the Japan Diabetes Society (JDS) value at the baseline, it was calculated using the following conversion equation from the JDS value to the National Glycohemoglobin Standardization Program (NGSP) value, according to the JDS statement [29]: HbA1c (%) = 1.02 × HbA1c (JDS) (%) + 0.25%. Based on the World Health Organization definition [30], anemia was defined as hemoglobin in males (adult) ≤ 13 g/dL and females (adult) ≤ 12 g/dL. Smoking status was categorized as nonsmoker, ex-smoker, daily smoker ≤ 20 cigarettes/day or >20 cigarettes/day, alcohol consumption was categorized as nondrinker, occasional drinker, drinker < 1 *go*/day or ≥1 *go*/day (“*go*” is the amount of sake: a traditional Japanese beverage, where 1 *go* contains approximately 23 g of ethanol), exercise was categorized as exercises causing light sweating exercises ≥30 min/time at frequency of ≥2 days/week, for ≥1 year, and job type was categorized as professional job, management, office job, sales, service, telecommunications, manufacturing, and other.

### 2.5. Statistical Analysis

Person-years were calculated from the day of the annual health check-up in the 2013 fiscal year (baseline) to the day of hearing impairment onset, or the day of the last annual health check-up, until March 2021 (whichever came first). A Cox proportional hazards model was applied to estimate hazard ratio (HR) and 95% confidence interval (CI) values for associations between eGFR and low- and high-frequency hearing impairment, relative to the lower half of the ≥90 mL/min/1.73 m^2^ group as the reference group. This analysis was performed according to participant sex because there are sex differences in renal and hearing impairments [31,32,33,34]. After exploration of crude and age-adjusted associations, a multivariable-adjusted model was constructed, included the following factors: hypertension, dyslipidemia, diabetes, BMI, smoking status, alcohol consumption, exercise, anemia, and job type. It was confirmed that the proportional hazards assumption was not violated in our models.

Several restricted analyses were performed as sensitivity analyses, to minimize the residual confounding, as follows: participants without diabetes, nonsmokers, those < 40 years old, those engaged in office jobs, and those with BMI 18.5–24.9 kg/m^2^. Further, to avoid overestimation of eGFR, based on low serum creatinine level related to low skeletal muscle mass, we excluded participants with serum creatinine levels below the standard value (male, <0.61 mg/dL; female, <0.47 mg/dL).

All statistical analyses were performed using Stata version 14.2 (StataCorp LLC, College Station, TX, USA).

## 3. Results

Participant baseline characteristics, stratified according to eGFR group, are shown in Table 1 (male) and Table 2 (female). The mean ages of male and female participants were 43.6 and 44.4 years, respectively, while mean eGFR values were 82.7 and 83.8 mL/min/1.73 m^2^, respectively. Mean follow-up period was approximately 5.3 years.

HR values for hearing impairment according to eGFR group are shown in Table 3 (male) and Table 4 (female). For males, eGFR < 60 mL/min/1.73 m^2^ was positively associated with low-frequency hearing impairment in the crude model; however, this association disappeared after adjustment for age. Compared with those in the lower half of the eGFR ≥ 90 mL/min/1.73 m^2^ group, the HR values for low-frequency hearing impairment were 0.84 (95% CI, 0.67–1.06) for eGFR 60–89 mL/min/1.73 m^2^ and 0.93 (95% CI, 0.63–1.36) for eGFR < 60 mL/min/1.73 m^2^ in the multivariable model. Similarly, there was no association between eGFR < 60 mL/min/1.73 m^2^ and high-frequency hearing impairment (multivariable-adjusted HR = 0.95, 95% CI, 0.79–1.14). Meanwhile, in the upper half of the ≥90 mL/min/1.73 m^2^ group, an elevated risk of high-frequency hearing impairment was observed (multivariable-adjusted HR = 1.16, 95% CI, 1.01–1.34); however, the association between high eGFR and high-frequency hearing impairment appeared not to be robust in several sensitivity analyses. Further, after exclusion of males with low creatinine level, the HR value was 1.14 (95% CI, 0.96–1.34). In females, eGFR appeared to increase the risk of low- and high-frequency hearing impairment in the crude model; however, after adjustment for several confounding factors, no association between eGFR and hearing impairment was detected.

HR values for hearing impairment related to covariates in the multivariable model are summarized in Appendix A. Age and job type, manufacturing (using office job as a reference) were positively associated with low- and high-hearing impairment in both sexes. Smoking, overweight (BMI ≥ 25.0 kg/m^2^), and anemia in males, and diabetes and underweight (BMI < 18.5 kg/m^2^) in females, were associated with increased risk of hearing impairment.

## 4. Discussion

Among the working-age population in Japan, there was no association of eGFR with low-frequency hearing impairment in either sex or with high-frequency hearing impairment in females. For males, although high eGFR appeared to be associated with an elevated risk of high-frequency hearing impairment, this positive association was not consistent in sensitivity analyses.

To our best knowledge, only one cohort study [35] has explored the association between eGFR and incidence of hearing impairment. In the Epidemiology of Hearing Loss study, which followed 1843 patients aged 48–80 years over 15 years, eGFR (calculated using the CKD-EPI formula) was not associated with hearing impairment. Compared with eGFR between 60 and 90 mL/min/1.73 m^2^, sex-combined multivariable-adjusted HR values for pure-tone average (PTA) > 25 dB hearing impairment at low frequencies (500, 1000, 2000 Hz) were 0.88 (95% CI, 0.63–1.22) for eGFR < 60 mL/min/1.73 m^2^ and 1.19 (95% CI, 0.92–1.53) for eGFR ≥ 90 mL/min/1.73 m^2^. Further, HR values for PTA > 25 dB hearing impairment at high frequencies (4000, 6000, 8000 Hz) were 0.90 (95% CI, 0.57–1.42) for eGFR < 60 mL/min/1.73 m^2^ and 1.02 (95% CI, 0.81–1.27) for eGFR ≥ 90 mL/min/1.73 m^2^. These results were broadly consistent with the findings of our study. Similar to our hypothesis, several, but not all [11,12,13,14], cross-sectional studies have reported that low eGFR increased the risk of hearing impairment; however, the results of these studies should be interpreted carefully because the eGFR estimation formula and hearing evaluation method used differed, and the temporality of the association was unclear.

In this study, high eGFR appeared to likely be associated with high-frequency hearing impairment in males; however, the majority of sensitivity analyses (Table 3), except analysis restricted to office job, displayed wide confidence intervals because of small sample size, resulting in small HR point estimates that approached null, indicating possible confounding effects on the association between eGFR and hearing impairment. These findings are also likely to be due to methodological issues associated with GFR estimation using serum creatinine levels [36]. Shafi et al. conducted a study to identify individual-level differences between measured GFR (mGFR), based on 24 h urine storage, as a gold standard for GFR measurement, and eGFR, in the US general population [37]. Differences between population-level mGFR and eGFR were small; however, differences between individual-level mGFR and eGFR were large, suggesting that eGFR does not correlate well with mGFR. Although we used equations for eGFR corrected for Japanese-specific factors, eGFR may have deviated from actual renal function. Another possibility is that high eGFR was a marker of low skeletal muscle mass, which is related to low serum creatinine level, as low skeletal muscle mass has previously been associated with hearing impairment [38,39]. In the present study, after excluding participants with serum creatinine level below the standard value (male, <0.61 mg/dL), the statistically significant association between high eGFR and high-frequency hearing impairment disappeared. Cystatin C-based eGFR is less susceptible to the influence of muscle mass [36]. If cystatin C data had been available, we may have explored the eGFR-hearing association following exclusion of the effect of muscle mass using cystatin C-based eGFR. Further investigation is required to assess whether the positive association between high eGFR and hearing impairment is genuine.

The strengths of this study include its longitudinal design, spanning a maximum of 8 years; use of large-scale occupational health check-up data from approximately 127,000 Japanese workers aged 20–59 years old. Nevertheless, this study has some limitations. First, because the evaluation of hearing impairment was only a pure-tone test of limited frequencies, outcome misclassification unrelated to exposure was likely to occur. Such misclassification is known to result in “bias towards the null”. However, established or suspected risk factors, including age, manufacturing work with possible noise exposure, diabetes, BMI, smoking, and anemia, were associated with hearing impairment in this study [7,8,9,21,23,24,40]. Thus, even if the outcome non-differential misclassification had masked the eGFR-hearing impairment association, it is less likely that renal dysfunction would have affected hearing more than the known risk factors for hearing impairment. Second, the number of participants with eGFR below 60 mL/min/1.73 m^2^ was low at approximately 3% for both males and females. Although most of signs and symptoms of CKD appears only when eGFR drops below 60 mL/min/1.73 m^2^, and several previous cross-sectional studies have reported that eGFR below 60 mL/min/1.73 m^2^ was associated with hearing impairment [11,13,14], we did not perform the analysis by the eGFR classification of CKD severity (45–59, 30–44, 15–29, <15 mL/min/1.73 m^2^). Third, information on occupational risk factors for hearing impairment, such as exposure to noise and injurious work, were not available in our study; however, to minimize the confounding effect of noise, we performed sensitivity analysis restricted to participants with office jobs. Finally, the healthy worker effect should be considered due to the fact that our population comprised exclusively individuals of working age [41].

## 5. Conclusions

eGFR is not generally associated with hearing impairment in either sex among the Japanese working-age population. To assess the tendency toward elevated risk of high-frequency hearing impairment among males in the highest eGFR group, further epidemiological studies of other populations will be required.

## Figures and Tables

**Figure 1 ijerph-19-12368-f001:**
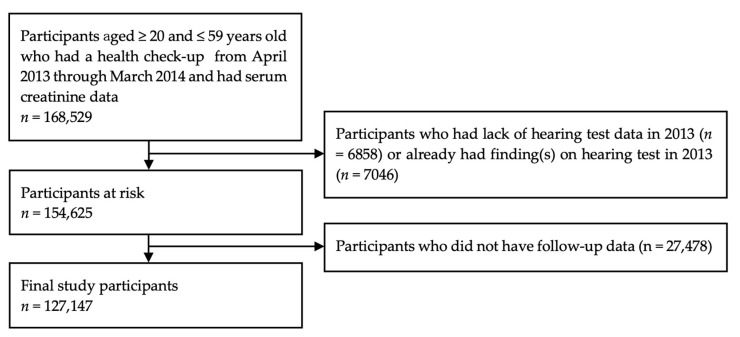
Participants flow.

**Table 1 ijerph-19-12368-t001:** Baseline characteristics according to estimated glomerular filtration rate groups in male participants.

		eGFR (mL/min/1.73 m^2^) Group			
	Overall	≥90		60–89	<60
		High (≥97.4)	Low (90.0–97.3)		
	n (%)	n (%)	n (%)	n (%)	n (%)
No. of participants	88,425	12,031	12,011	61,459	2924
Age (years) ^a^	43.6 (8.7)	37.8 (9.0)	41.5 (9.2)	44.7 (8.0)	50.5 (6.0)
20s	6088 (6.9)	2349 (19.5)	1449 (12.1)	2286 (3.7)	4 (0.2)
30s	22,097 (25.0)	4670 (38.8)	3413 (28.4)	13,899 (22.6)	115 (3.9)
40s	35,866 (40.6)	3635 (30.2)	4481 (37.3)	26,679 (43.4)	1071 (36.6)
50s	24,374 (27.5)	1377 (11.5)	2668 (22.2)	18,595 (30.3)	1734 (59.3)
eGFR ^a^	82.7 (14.0)	106.7 (9.4)	93.5 (2.2)	77.2 (7.6)	54.4 (6.3)
Body mass index (kg/m^2^) ^a^	23.6 (3.7)	23.1 (4.1)	23.2 (3.7)	23.7 (3.5)	25.0 (3.7)
<18.5	4050 (4.6)	972 (8.1)	756 (6.3)	2277 (3.7)	45 (1.5)
18.5–24.9	58,389 (66.0)	8031 (66.7)	8187 (68.2)	40,611 (66.1)	1560 (53.4)
≥25.0	25,986 (29.4)	3028 (25.2)	3068 (25.5)	18,571 (30.2)	1319 (45.1)
Hypertension ^b^	41,139 (46.5)	4884 (40.6)	5198 (43.3)	29,093 (47.3)	1964 (67.2)
Dyslipidemia ^c^	27,140 (30.7)	3196 (26.6)	3289 (27.4)	19,340 (31.5)	1315 (45.0)
Diabetes ^d^	7545 (8.5)	1151 (9.6)	1090 (9.1)	4870 (7.9)	434 (14.8)
Hemoglobin A1c (%) ^a^	5.8 (0.7)	5.8 (1.0)	5.8 (0.8)	5.8 (0.6)	6.0 (0.7)
Smoking status					
Nonsmoker	32,078 (36.3)	3832 (31.8)	3816 (31.8)	23,052 (37.5)	1378 (47.1)
Ex-smoker	14,509 (16.4)	1368 (11.4)	1763 (14.7)	10,725 (17.5)	653 (22.3)
≤20 cigarettes/day	30,205 (34.2)	4906 (40.8)	4616 (38.4)	20,050 (32.6)	633 (21.7)
>20 cigarettes/day	9118 (10.3)	1632 (13.6)	1478 (12.3)	5841 (9.5)	167 (5.7)
Unanswered	2515 (2.8)	293 (2.4)	338 (2.8)	1791 (2.9)	93 (3.2)
Alcohol consumption					
Nondrinker	24,489 (27.7)	3440 (28.6)	3146 (26.2)	16,975 (27.6)	928 (31.7)
Occasional drinker	30,712 (34.7)	4404 (36.6)	4212 (35.1)	21,067 (34.3)	1029 (35.2)
Drinker <1 *go*/day ^e^	7906 (8.9)	880 (7.3)	988 (8.2)	5779 (9.4)	259 (8.9)
Drinker ≥1 *go*/day ^e^	22,798 (25.8)	3018 (25.1)	3322 (27.7)	15,843 (25.8)	615 (21.0)
Unanswered	2520 (2.9)	289 (2.4)	343 (2.8)	1795 (2.9)	93 (3.2)
Exercise, yes ^f^	17,652 (20.0)	2155 (17.9)	2248 (18.7)	12,596 (20.5)	653 (22.3)
Anemia ^g^	1414 (1.6)	210 (1.8)	155 (1.3)	919 (1.5)	130 (4.5)
Job type					
Professional job	22,706 (25.7)	3259 (27.1)	3078 (25.6)	15,685 (25.5)	684 (23.4)
Management	10,962 (12.4)	911 (7.6)	1233 (10.3)	8284 (13.5)	534 (18.3)
Office job	11,067 (12.5)	1244 (10.3)	1385 (11.5)	7968 (13.0)	470 (16.1)
Sales	7661 (8.7)	1302 (10.8)	1123 (9.4)	5037 (8.2)	199 (6.8)
Service	3785 (4.3)	687 (5.7)	570 (4.8)	2453 (4.0)	75 (2.6)
Telecommunications	6125 (6.9)	772 (6.4)	829 (6.9)	4301 (7.0)	223 (7.6)
Manufacturing	18,182 (20.6)	2766 (23.0)	2695 (22.4)	12,246 (19.9)	475 (16.2)
Other	5265 (5.9)	767 (6.4)	736 (6.1)	3595 (5.8)	167 (5.7)
Unanswered	2672 (3.0)	323 (2.7)	362 (3.0)	1890 (3.1)	97 (3.3)

Notes: ^a^ Mean (SD). ^b^ Systolic blood pressure ≥ 130 mm Hg, diastolic blood pressure ≥ 85 mm Hg, or receiving medication. ^c^ Triglyceride level ≥ 150 mg/dL (1.7 mmol /L), high-density lipoprotein cholesterol level < 40 mg/dL (1.03 mmol /L), or receiving medication. ^d^ Self-reported diagnosis of diabetes, receiving medication, or HbA1c ≥ 6.5%. ^e^ One *go* contains approximately 23 g of ethanol. ^f^ Exercise leading to light sweating ≥ 30 min/time (frequency, ≥2 days/week; duration, ≥1 year). ^g^ Hemoglobin (g/dL) ≤ 13 g/dL. Abbreviation: eGFR, estimated glomerular filtration rate.

**Table 2 ijerph-19-12368-t002:** Baseline characteristics according to estimated glomerular filtration rate groups in female participants.

		eGFR (mL/min/1.73 m^2^) Groups			
	Overall	≥90		60–89	<60
		High (≥99.6)	Low (90.0–99.5)		
	n (%)	n (%)	n (%)	n (%)	n (%)
No. of participants	38,722	5743	5593	26,005	1381
Age (years) ^a^	44.4 (9.1)	38.1 (9.1)	41.3 (10.2)	46.1 (8.0)	51.6 (5.7)
20s	2883 (7.5)	1184 (20.6)	825 (14.8)	871 (3.3)	3 (0.2)
30s	8013 (20.7)	1862 (32.4)	1714 (30.6)	4385 (16.9)	52 (3.8)
40s	15,036 (38.8)	2113 (36.8)	1528 (27.3)	11,049 (42.5)	346 (25.0)
50s	12,790 (33.0)	584 (10.2)	1526 (27.3)	9700 (37.3)	980 (71.0)
eGFR ^a^	83.8 (15.3)	110.0 (10.9)	94.6 (2.8)	77.2 (7.6)	55.4 (5.6)
Body mass index (kg/m^2^) ^a^	21.9 (3.8)	21.7 (3.9)	21.7 (3.8)	22.0 (3.8)	22.7 (4.0)
<18.5	5576 (14.4)	935 (16.3)	926 (16.6)	3581 (13.8)	134 (9.7)
18.5–24.9	26,341 (68.0)	3886 (67.7)	3788 (67.7)	17,759 (68.3)	908 (65.7)
≥25.0	6805 (17.6)	922 (16.0)	879 (15.7)	4665 (17.9)	339 (24.6)
Hypertension ^b^	11,034 (28.5)	1230 (21.4)	1382 (24.7)	7800 (30.0)	622 (45.0)
Dyslipidemia ^c^	6267 (16.2)	857 (14.9)	829 (14.8)	4238 (16.3)	343 (24.8)
Diabetes ^d^	2116 (5.5)	377 (6.6)	327 (5.9)	1309 (5.0)	103 (7.5)
Hemoglobin A1c (%) ^a^	5.7 (0.5)	5.7 (0.8)	5.7 (0.6)	5.7 (0.4)	5.9 (0.6)
Smoking status					
Nonsmoker	28,360 (73.2)	3971 (69.2)	3981 (71.2)	19,321 (74.3)	1087 (78.7)
Ex-smoker	2630 (6.8)	453 (7.9)	384 (6.9)	1705 (6.6)	88 (6.4)
≤20 cigarettes/day	6341 (16.4)	1071 (18.6)	985 (17.6)	4116 (15.8)	169 (12.2)
>20 cigarettes/day	411 (1.1)	79 (1.4)	65 (1.1)	253 (1.0)	14 (1.0)
Unanswered	980 (2.5)	169 (2.9)	178 (3.2)	610 (2.3)	23 (1.7)
Alcohol consumption					
Nondrinker	20,024 (51.7)	2833 (49.3)	2737 (48.9)	13,667 (52.6)	787 (57.0)
Occasional drinker	12,591 (32.5)	1988 (34.6)	1979 (35.4)	8226 (31.6)	398 (28.8)
Drinker <1 *go*/day ^e^	2317 (6.0)	303 (5.3)	298 (5.3)	1621 (6.2)	95 (6.9)
Drinker ≥1 *go*/day ^e^	2810 (7.3)	451 (7.9)	400 (7.2)	1881 (7.2)	78 (5.6)
Unanswered	980 (2.5)	168 (2.9)	179 (3.2)	610 (2.4)	23 (1.7)
Exercise, yes ^f^	4724 (12.2)	530 (9.2)	489 (8.7)	3406 (13.1)	299 (21.7)
Anemia ^g^	7022 (18.1)	1403 (24.4)	1026 (18.3)	4430 (17.0)	163 (11.8)
Job type					
Professional job	2731 (7.0)	450 (7.8)	466 (8.3)	1744 (6.7)	71 (5.1)
Management	688 (1.8)	101 (1.8)	101 (1.8)	462 (1.8)	24 (1.7)
Office job	14,755 (38.1)	2216 (38.6)	1967 (35.2)	10,011 (38.5)	561 (40.6)
Sales	2720 (7.0)	579 (10.1)	464 (8.3)	1592 (6.1)	85 (6.2)
Service	2579 (6.7)	407 (7.1)	433 (7.7)	1669 (6.4)	70 (5.1)
Telecommunications	293 (0.8)	44 (0.8)	34 (0.6)	206 (0.8)	9 (0.7)
Manufacturing	9503 (24.5)	1156 (20.1)	1375 (24.6)	6624 (25.5)	348 (25.2)
Other	4267 (11.0)	577 (10.0)	548 (9.8)	2958 (11.4)	184 (13.3)
Unanswered	1186 (3.1)	213 (3.7)	205 (3.7)	739 (2.8)	29 (2.1)

Notes: ^a^ Mean (SD). ^b^ Systolic blood pressure ≥ 130 mm Hg, diastolic blood pressure ≥ 85 mm Hg, or receiving medication. ^c^ Triglyceride level ≥ 150 mg/dL (1.7 mmol /L), high-density lipoprotein cholesterol level < 50 mg/dL (1.3 mmol/L), or receiving medication. ^d^ Self-reported diagnosis of diabetes, receiving medication, or HbA1c ≥ 6.5%. ^e^ One *go* contains approximately 23 g of ethanol. ^f^ Exercise leading to light sweating ≥ 30 min/time (frequency, ≥2 days/week; duration, ≥1 year). ^g^ Hemoglobin (g/dL) ≤ 12 g/dL. Abbreviation: eGFR, estimated glomerular filtration rate.

**Table 3 ijerph-19-12368-t003:** Hearing impairment according to estimated glomerular filtration rate groups in male participants.

	eGFR (mL/min/1.73 m^2^) Groups			
	≥90		60–89	<60
	High (≥97.4)	Low (90.0–97.3)		
Low frequency (1000 Hz)				
Person-years	63,645	65,077	338,372	15,717
No. of cases	66	94	502	39
Crude HR (95% CI)	0.72 (0.53–0.99)	Ref.	1.02 (0.82–1.27)	1.71 (1.18–2.49)
Age-adjusted HR (95% CI)	1.05 (0.77–1.44)	Ref.	0.81 (0.65–1.01)	0.85 (0.58–1.24)
Multivariable-adjusted HR (95% CI) ^a^	1.01 (0.73–1.39)	Ref.	0.84 (0.67–1.06)	0.93 (0.63–1.36)
Restricted to males without diabetes	1.05 (0.75–1.48)	Ref.	0.84 (0.66–1.07)	0.86 (0.56–1.33)
Restricted to nonsmokers	0.70 (0.37–1.35)	Ref.	0.65 (0.44–0.96)	0.74 (0.41–1.35)
Restricted to males aged <40 y.o.	1.39 (0.63–3.08)	Ref.	1.04 (0.50–2.19)	3.27 (0.40–26.41)
Restricted to office job	0.69 (0.21–2.21)	Ref.	0.64 (0.32–1.26)	0.41 (0.11–1.52)
Restricted to males with body mass index 18.5–24.9 kg/m^2^	1.10 (0.75–1.61)	Ref.	0.85 (0.65–1.12)	0.79 (0.47–1.36)
Excluded males with low creatinine level ^b^	0.94 (0.64–1.39)	Ref.	0.85 (0.68–1.06)	0.93 (0.64–1.37)
High frequency (4000 Hz)				
Person-years	62,814	64,047	332,901	15,383
No. of cases	366	443	2379	161
Crude HR (95% CI)	0.84 (0.73–0.97)	Ref.	1.03 (0.93–1.14)	1.51 (1.26–1.81)
Age-adjusted HR (95% CI)	1.22 (1.07–1.41)	Ref.	0.82 (0.74–0.91)	0.76 (0.63–0.91)
Multivariable-adjusted HR (95% CI) ^a^	1.16 (1.01–1.34)	Ref.	0.91 (0.82–1.01)	0.95 (0.79–1.14)
Restricted to males without diabetes	1.15 (0.99–1.34)	Ref.	0.90 (0.81–1.00)	0.90 (0.73–1.10)
Restricted to nonsmokers	0.95 (0.69–1.31)	Ref.	0.80 (0.65–0.98)	0.86 (0.63–1.18)
Restricted to males aged <40 y.o.	0.89 (0.62–1.28)	Ref.	0.74 (0.53–1.02)	0.41 (0.06–2.97)
Restricted to office job	1.48 (0.78–2.81)	Ref.	0.95 (0.59–1.51)	0.94 (0.46–1.91)
Restricted to males with body mass index 18.5–24.9 kg/m^2^	1.04 (0.88–1.23)	Ref.	0.83 (0.74–0.94)	0.88 (0.69–1.12)
Excluded males with low creatinine level ^b^	1.14 (0.96–1.34)	Ref.	0.91 (0.82–1.01)	0.95 (0.79–1.15)

Notes: ^a^ Adjusted for age, hypertension, dyslipidemia, diabetes, body mass index, smoking status, alcohol consumption, exercise, anemia, and job type. ^b^ Excluding participants with serum creatinine <0.61 mg/dL. Abbreviations: CI, confidence interval; eGFR, estimated glomerular filtration rate; HR, hazard ratio; y.o., years old.

**Table 4 ijerph-19-12368-t004:** Hearing impairment according to estimated glomerular filtration rate group in female participants.

	eGFR (mL/min/1.73 m^2^) Group			
	≥90		60–89	<60
	High (≥99.6)	Low (90.0–99.5)		
Low frequency (1000 Hz)				
Person-years	28,928	28,916	138,184	7270
No. of cases	39	61	307	18
Crude HR (95% CI)	0.63 (0.42–0.94)	Ref.	1.04 (0.79–1.37)	1.17 (0.69–1.98)
Age-adjusted HR (95% CI)	0.94 (0.63–1.41)	Ref.	0.83 (0.63–1.10)	0.63 (0.37–1.07)
Multivariable-adjusted HR (95% CI) ^a^	0.90 (0.60–1.36)	Ref.	0.85 (0.64–1.12)	0.66 (0.39–1.12)
Restricted to females without diabetes	0.99 (0.63–1.54)	Ref.	0.93 (0.68–1.25)	0.82 (0.48–1.42)
Restricted to nonsmokers	0.98 (0.61–1.58)	Ref.	0.88 (0.64–1.22)	0.75 (0.42–1.33)
Restricted to females aged <40 y.o.	0.77 (0.33–1.79)	Ref.	0.81 (0.38–1.72)	– (No event)
Restricted to office job	0.73 (0.32–1.69)	Ref.	0.79 (0.46–1.34)	0.37 (0.12–1.11)
Restricted to females with body mass index 18.5–24.9 kg/m^2^	0.94 (0.56–1.58)	Ref.	0.83 (0.58–1.17)	0.75 (0.39–1.44)
Excluded females with low creatinine level ^b^	0.88 (0.55–1.42)	Ref.	0.85 (0.65–1.12)	0.67 (0.39–1.14)
High frequency (4000 Hz)				
Person-years	28,968	28,963	138,250	7247
No. of cases	27	37	247	19
Crude HR (95% CI)	0.73 (0.44–1.19)	Ref.	1.39 (0.98–1.96)	2.05 (1.18–3.55)
Age-adjusted HR (95% CI)	1.28 (0.77–2.12)	Ref.	1.08 (0.77–1.53)	0.98 (0.56–1.71)
Multivariable-adjusted HR (95% CI) ^a^	1.23 (0.75–2.04)	Ref.	1.13 (0.80–1.60)	1.06 (0.61–1.85)
Restricted to females without diabetes	1.36 (0.80–2.31)	Ref.	1.11 (0.77–1.61)	0.73 (0.38–1.43)
Restricted to nonsmokers	1.31 (0.73–2.35)	Ref.	1.10 (0.73–1.65)	1.20 (0.65–2.22)
Restricted to females aged <40 y.o.	1.76 (0.44–7.09)	Ref.	0.68 (0.16–2.78)	– (No event)
Restricted to office job	0.69 (0.23–2.05)	Ref.	0.91 (0.47–1.78)	0.14 (0.02–1.13)
Restricted to females with body mass index 18.5–24.9 kg/m^2^	0.94 (0.50–1.78)	Ref.	0.97 (0.65–1.46)	0.61 (0.28–1.35)
Excluded females with low creatinine level ^b^	1.02 (0.54–1.96)	Ref.	1.13 (0.80–1.60)	1.05 (0.60–1.83)

Notes: ^a^ Adjusted for age, hypertension, dyslipidemia, diabetes, body mass index, smoking status, alcohol consumption, exercise, anemia, and job type. ^b^ Excluding participants with serum creatinine <0.47 mg/dL. Abbreviations: CI, confidence interval; eGFR, estimated glomerular filtration rate; HR, hazard ratio; y.o., years old.

## Data Availability

The data are not publicly available due to ethical considerations.

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
