# Peer review of "Estimated Glomerular Filtration Rate and Hearing Impairment in Japan: A Longitudinal Analysis Using Large-Scale Occupational Health Check-Up Data"

_ijerph, 2022, doi:10.3390/ijerph191912368_

Round 1
Reviewer 1 Report
This study is related to estimated GFR and hearing impairment among the working age population in Japan.
This is a longitudinal analysis that included large number of males and females, but I have some comments and questions:
I do not fully understand the main intention of your research. In this study, you hypothesized the renal dysfunction can lead to hearing impairment and investigated the longitudinal association between eGFR, as a marker of renal function and hearing impairment. But the working population that you investigated was divided into four groups, according to the eGFR. Only a few participants had eGFR < 60 (Table 1 and Table 2). Most of signs and simptoms of CKD (including possible hearing impairment) appears only when eGFR drops below 60. This can also affect final results.
In cases when you found hearing impairment, did you analyzed if there were any changes in the values of eGFR before impairment appeared and at the moment it began.
At the end of discussion, you stated:..."outcome misclassification unrelated to exposure was likely to occur, but would not be expected to have a substantial impact on eGFR-hearing impairment association..." I am not convinced that this is so. Would you, please, explain it clearly, why do you think so?
Author Response
Thank you very much for reviewing our manuscript. I have summarized the reply in a Word file, so please check the attached file.

Reviewer 2 Report
The paper, “Estimated glomerular filtration rate and hearing impairment in Japan: a longitudinal analysis using large-scale occupational health check-up data” has been written well and will add value to the special edition of the journal. There are few recommendations which may make it better.
1. First few lines of the introduction are not impressive, these should be re-written, for example; second line should be first and first line should be second.
2. Authors have written reference numbers like [11,12,13,14], which should be [11-14] so the style of citing the reference should be revised in the whole text of the manuscript.
3. In the section 2.1 Study population, the selection process of the sample is little confusing. The authors are advised to add a flow chart clearly mentioning inclusion/exclusion criteria.
4. All baseline variables must be defined properly in a separate para under the title, “Definition of baseline variables”. The cutoffs must be checked especially for BMI and Hypertension as the values for hypertension mentioned by the authors are different from the Japanese Society of Hypertension Guidelines for the Management of Hypertension (PMID: 31375757).
5. The 255th line in the third paragraph, mentioned that young people in their 20s are involved in the study. Clarify it, as the age range of the subjects taken is different.
Author Response

(The authors gave the same response as above.)

Round 2
Reviewer 1 Report
I thank the authors for the response. I don t have any further questions.